# Knowledge about mother to child transmission of HIV/AIDS, its prevention and associated factors among reproductive-age women in sub-Saharan Africa: Evidence from 33 countries recent Demographic and Health Surveys

Achamyeleh Birhanu Teshale[1]*, Zemenu Tadesse Tessema[1], Adugnaw Zeleke Alem[1], Yigizie Yeshaw[1,2], Alemneh Mekuriaw Liyew[1], Tesfa Sewunet Alamneh[1], Getayeneh Antehunegn Tesema[1], Misganaw Gebrie Worku[3]

1 Department of Epidemiology and Biostatistics, Institute of Public Health, College of Medicine and Health Sciences, University of Gondar, Gondar, Ethiopia, 2 Department of Physiology, School of Medicine, College of Medicine and Health Sciences, University of Gondar, Gondar, Ethiopia, 3 Department of Human Anatomy, College of Medicine and Health Science, School of Medicine, University of Gondar, Gondar, Ethiopia

* achambir08@gmail.com

## Abstract

### Background

In sub-Saharan Africa (SSA) 90 percent of babies acquired HIV/AIDS from infected mothers. Maternal knowledge about mother to child transmission (MTCT) of HIV/AIDS and its prevention is a cornerstone for elimination of MTCT of HIV/AIDS. Despite this, there is limited evidence about knowledge about MTCT of HIV/AIDS and its prevention and associated factors in SSA. Therefore, this study aimed to assess knowledge of MTCT of HIV/AIDS, its prevention (PMTCT) and, associated factors among reproductive-age women in sub-Saharan Africa.

### Objective

To assess Knowledge about mother to child transmission of HIV/AIDS and its prevention and associated factors among reproductive-age women in Sub-Saharan Africa.

### Methods

The recent SSA countries' Demographic and Health Surveys (DHS), which were conducted from 2008/09 to 2018/19, was our data source. We appended 33 countries' DHS data for our analysis. For our study, a total weighted sample of 350,888 reproductive-age women was used. Due to the hierarchical nature of the DHS data, we conducted a multilevel analysis. Finally, the adjusted odds ratio with its 95% confidence interval was reported, and

**Data Availability Statement:** All relevant data are within the paper and its Supporting Information files.

**Funding:** The author(s) received no specific funding for this work.

**Competing interests:** The authors have declared that no competing interests exist.

**Abbreviations:** AIDS, Acquired Immune Deficiency Syndrome; AOR, Adjusted Odds Ratio; CI, Confidence Interval; DHS, Demographic and Health Surveys; HIV, Human Immune Deficiency Virus; ICC, Intra-class Correlation Coefficient; MTCT, Mother to child Transmission; PMTCT, Prevention of Mother to Child Transmission.

variables with p-value≤0.05 were considered as significant predictors of knowledge of MTCT of HIV/AIDS and its prevention.

## Results

In this study, 56.21% (95% CI: 56.05–56.38) of respondents had correct knowledge about MTCT of HIV/AIDS and its prevention among reproductive-age women in SSA. In the multilevel logistic regression analysis: being in the older age group, better education level, being from a rich household, having mass media exposure, having parity of one and above were associated with higher odds of knowledge about MTCT of HIV/AIDS and its prevention. However, being perceiving distance from the health facility as a big problem was associated with lower odds of knowledge about MTCT of HIV/AIDS and its prevention.

## Conclusion

Knowledge about MTCT of HIV/AIDS and its prevention among reproductive-age women in SSA was low. Therefore, it is better to consider the high-risk groups during the intervention to increase awareness about this essential public health issue and to tackle its devastating outcome.

## Background

Morbidity and mortality due to human immunodeficiency virus (HIV) infection have decreased worldwide over the past decade due to preventive programs such as increased coverage of antiretroviral therapy (ART) and prevention of mother-to-child transmission (MTCT) of HIV/AIDS [1]. Prevention of mother-to-child transmission (PMTCT) program has prevented approximately 1.4 million new childhood HIV infections and is a major contributor to the elimination of new HIV infections in low- and middle-income countries [2,3].

In the era of Option B+, initiation of antiretroviral therapy for all pregnant mothers to PMTCT of HIV/AIDS, a larger number of women living with HIV (WLHIV) are on ART and while more than 70% of WLHIV are on ART during pregnancy in most SSA countries. However, there are still gaps to improve uptake and adherence of ART [3]. Early intiation ART can suppress maternal viral load, and each additional week of ART during the antenatal period will reduce MTCT of HIV by 20% [4–6]. About 50% of the 180,000 new pediatric HIV infections in 2017 were infected during breastfeeding and it is estimated that in the absence of any intervention to prevent MTCT, the risk of transmission ranges from 15–45 percent (5–10 percent during pregnancy, 10–20 percent during childbirth, and 10–20 percent via mixed infant feeding) [7]. However, with successful measures, this rate can be decreased to less than 5% [7].

Nearly 90 percent of all children and adolescents living with HIV are in Sub-Saharan Africa (SSA) [8]. Yet, the effect of the epidemic among those populations varies widely throughout the region [8]. Despite substantial improvements in the accessibility of ART, around 7.9 percent of children died in SSA countries [9]. Even though 90 percent of babies who acquire the disease from infected mothers are found in SSA [10], maternal knowledge about MTCT of HIV/AIDS and its prevention is low ranging from 34.9 percent in Ethiopia to 78 percent in Nigeria [11–16].

According to various studies done elsewhere, knowledge about MTCT and PMTCT of HIV/AIDS is correlated with factors such as maternal age, maternal education, wealth status,

occupation, marital status, media exposure, and residence [11,12,17–19]. Maternal knowledge about MTCT of HIV/AIDS and its prevention is a cornerstone for elimination of MTCT of HIV. Although the majority of the population in SSA are lived in rural areas with restricted availability and accessibility of health facilities, most of the studies on knowledge about MTCT of HIV/AIDS and its prevention were conducted among available women, such as those who came to the health facility for their antenatal care follow up [20–24]. In addition, up to our knowledge, there is no updated information on this regard using nationally representative data (using the recent DHS surveys) at the SSA scale. Therefore, this study aimed to assess knowledge of MTCT of HIV/AIDS, its prevention (PMTCT), and associated factors among reproductive-age women in 33 sub-Saharan Africa countries.

## Methods

### Data source and study population

The recent SSA countries Demographic and Health Surveys (DHS), which were conducted from 2008/09 to 2018/19, was our data source. There were 35 countries DHS conducted in the study period. However, we appended 33 countries' DHS data for our analysis since the two countries (Senegal and Tanzania) DHS had no observation regarding our outcome variable. For our study, a total weighted sample of 350,888 reproductive-age women was used.

### Study variables

**Outcome variable.** The outcome variable in this study was knowledge about MTCT of HIV/AIDS and its prevention (PMTCT). It was a composite score of four different questions: HIV can be transmitted from a mother to her baby during pregnancy (yes/no), HIV be transmitted from a mother to her baby during delivery (yes/no), HIV be transmitted from a mother to her baby by breastfeeding (yes/no), there are special medicines that a doctor or a nurse can give to a woman infected with HIV to reduce the risk of transmission to the baby (yes/no).

Then a woman had correct knowledge if she answers all the four questions correctly (if the woman said yes for all of the questions) and not knowledgeable if she did not give the correct answer for at least one of the questions.

**Independent variables.** Both individual level and community level independent variables were incorporated in assessing factors associated with knowledge about MTCT of HIV/AIDS and its prevention among reproductive-age women in SSA.

**Individual-level variables:** maternal age, maternal education, current marital status, household wealth status, religion, employment, mass media exposure (television, newspaper, and/or radio), parity, number of under-five children, and distance from the health facility were incorporated as individual-level factors.

**Community-level variables:** Residence, community illiteracy level, and community level of non-media exposure were the community-level variables. The two community-level variables (community illiteracy level and community level of non-media exposure) were created by aggregating the individual-level women education and media exposure, respectively, at the cluster level. These community-level variables were categorized as low and high based on the national median value.

### Data management and statistical analysis

We extract, recode, and do the analysis using Stata version 14 software. Throughout the analysis, weighting was done to assure the representativeness and non-response rate as well as to get an appropriate statistical estimate (robust standard error) [25]. Due to the hierarchical nature

of the DHS data, we conducted a multilevel analysis. In conducting the analysis, we fit four models. The first model (Null model) was fitted without any explanatory variables to assess the variability of the outcome between clusters or to assess the intra-class correlation coefficient (ICC). The second model (Model 2) was done by incorporating individual-level variables only. The third (Model 3) and the fourth (Model 4) models were fitted with community level only and both individual and community level variables respectively. For random effect analysis (to assess the community or cluster level variability of comprehensive knowledge about MTCT and its prevention, ICC and proportional change in variance (PCV) were used. Deviance was used to verify model fitness, and the best-fit model has been deemed a model with the lowest deviance.

To select eligible variables for the multivariable analysis we conducted a bivariable analysis. Those variables with a p-value less than or equal to 0.20 in the bivariable analysis were eligible for the multivariable analysis. Then, in the multivariable analysis, the adjusted odds ratio (AOR) with its 95% confidence interval (CI) was reported, and variables with p-value≤0.05 were considered as significant predictors of knowledge about MTCT of HIV/AIDS and its prevention.

### Ethical consideration

Since we were using publicly accessible data, ethical approval was not needed. In addition, this research was considered exempt by the Institute of Public Health, College of Medicine and Health Sciences, University of Gondar Institutional Review Committee. However, by registering or online requesting we have accessed the data set from the DHS website (**https:/ dhsprogram.com**).

## Results

### Sociodemographic characteristic of respondents

For the final analysis, we used a total weighted sample of 350,888 reproductive-age women. Most of the study participants (9.90%) were from Nigeria (S1 Table). The median age of participants was 28 years with IQR = 21–36 years. The majority (35.78%) of respondents had attended secondary education. Around two-thirds (63.36%) of respondents were currently married and 60.56% were employed. The majority (64.07%) of respondents did not perceive distance from the health facility as a big problem and around three-fourth (73.66%) of the participants had exposure to at least one media (television, newspaper, and/or radio). Regarding place of residence, most (58.37%) of respondents were from rural areas (Table 1).

### Knowledge about MTCT of HIV/AIDS and its prevention among reproductive-age women in SSA

Among 350,888 participants, all of them ever heard about HIV/AIDS. About 81.56%, 86.82%, 89.66%, and 78.52% of participants know that HIV can be transmitted during pregnancy, delivery, breastfeeding, and know that there are certain medications to prevent MTCT of HIV/AIDS respectively. However, 56.21% (95% CI: 56.05–56.38) of respondents had correct knowledge about MTCT of HIV/AIDS with great variation between countries ranging from 13.56% in Comoros to 76.02% in Zambia (Tables 2 and S2).

### Factors associated with knowledge about MTCT of HIV/AIDS and its prevention in SSA

**Random effect analysis.**    Table 3 revealed the random effect model and model comparison. In the null model, the ICC was 23%, which showed that 23% of the variation on

**Table 1. Sociodemographic characteristics of respondents.**

| Variables | Number of respondents (N = 350888) | Percentage (%) |
|---|---|---|
| Maternal age | | |
| 15–19 | 68959 | 19.65 |
| 20–24 | 66145 | 18.85 |
| 25–29 | 63156 | 18.00 |
| 30–34 | 52412 | 14.94 |
| 35–39 | 43311 | 12.34 |
| 40–44 | 31518 | 8.98 |
| 45–49 | 25387 | 7.23 |
| Maternal education | | |
| No formal education | 92418 | 26.34 |
| Primary education | 111717 | 31.84 |
| Secondary education | 125535 | 35.78 |
| Tertiary and higher education | 21218 | 6.05 |
| Current marital status | | |
| Married | 222308 | 63.36 |
| Not married | 128580 | 36.64 |
| Employment | | |
| Employed | 212486 | 60.56 |
| Not employed | 138402 | 39.44 |
| Wealth index | | |
| Poorest | 56010 | 15.96 |
| Poorer | 62152 | 17.71 |
| Middle | 66853 | 19.05 |
| Richer | 76165 | 21.71 |
| Richest | 89708 | 25.57 |
| Mass media exposure | | |
| Had exposure | 258456 | 73.66 |
| Not had exposure | 92432 | 26.34 |
| Distance from the health facility | | |
| A big problem | 126076 | 35.93 |
| Not a big problem | 224812 | 64.07 |
| Parity | | |
| Nulliparous | 91023 | 25.94 |
| Primiparous | 52144 | 14.86 |
| Multiparous | 123281 | 35.13 |
| Grand multiparous | 84440 | 24.06 |
| Number of under-five children | | |
| None | 106337 | 30.31 |
| 1–2 | 199719 | 56.92 |
| 3–6 | 44832 | 12.78 |
| Residence | | |
| Urban | 146083 | 41.63 |
| Rural | 204805 | 58.37 |
| Community illiteracy level | | |
| Low | 178650 | 50.91 |
| High | 172238 | 49.09 |
| Community-level of media non-exposure | | |

(*Continued*)

**Table 1.** (Continued)

| Variables | Number of respondents (N = 350888) | Percentage (%) |
|---|---|---|
| Low | 154130 | 43.93 |
| High | 196758 | 56.07 |

knowledge about MTCT of HIV/AIDS and its prevention in SSA was attributed due to differences between clusters or communities. In addition, the proportional change in variance (PCV) in the final model revealed that about 38.78% of the variation of knowledge about MTCT of HIV/AIDS and its prevention in SSA was explained by both individual and community-level factors. Regarding model comparison, the fourth model (Model 3) was the best-fitted model since it had the lowest deviance (468,878) (Table 3).

**Fixed effect analysis.** In the bivariable analysis, all factors were eligible (had $p \leq 0.20$) for the multivariable analysis. In the multivariable multilevel analysis, individual-level factors: maternal age, maternal education, household wealth status, mass media exposure, distance from the health facility, and parity were found to be significant factors associated with knowledge about MTCT of HIV/AIDS and its prevention among reproductive-age women in SSA. The odds of having knowledge about MTCT of HIV/AIDS and its prevention was higher among older women as compared to women aged from 15–19 years. The odds of having knowledge about MTCT of HIV/AIDS and its prevention was 1.22 (AOR = 1.22; 95%CI: 1.20–1.25), 1.35 (AOR = 1.35; 95%CI: 1.33–1.38), and 1.41 (AOR = 1.41; 95%CI: 1.36–1.46) times higher among mothers who had primary education, secondary education, and tertiary and higher education respectively as compared to those who did not attend formal education. Mothers from poor, middle, richer, and richest households had 1.07 (AOR = 1.07; 95%CI: 1.05–1.10), 1.15 (AOR = 1.15; 95%CI: 1.12–1.17), 1.22 (AOR = 1.22; 95%CI: 1.19–1.25), and 1.25 (AOR = 1.25; 95%CI: 1.22–1.29) times higher odds of knowledge about MTCT of HIV/AIDS and its prevention as compared to those who were from the poorest household. Mothers who had mass media exposure had 1.10 (AOR = 1.10; 95%CI: 1.09–1.12) times higher odds of knowledge about MTCT of HIV/AIDS and its prevention as compared to their counterparts. Mothers who perceive distance from the health facility as a big problem had 4% (AOR = 0.96;

**Table 2. Knowledge about MTCT of HIV/AIDS and its prevention among reproductive-age women in SSA.**

| Knowledge of MTCT and PMTCT | Number of respondents (N = 12763) | Percentage (%) |
|---|---|---|
| 1. HIV transmitted during pregnancy | | |
| No | 64697 | 18.44 |
| Yes | 286191 | 81.56 |
| 2. HIV transmitted during delivery | | |
| No | 46261 | 13.18 |
| Yes | 304627 | 86.82 |
| 3. HIV transmitted during breastfeeding | | |
| No | 36284 | 10.34 |
| Yes | 314604 | 89.66 |
| 4. There are drugs to avoid HIV transmission to the child during pregnancy (PMTCT) | | |
| No | 75372 | 21.48 |
| Yes | 275516 | 78.52 |
| 5. Comprehensive knowledge of MTCT and PMTCT | | |
| Knowledgeable | 197240 | 56.21 |
| Not knowledgeable | 153648 | 43.79 |

**Table 3. Random effect analysis and model comparison in the assessment of factors associated with knowledge about MTCT of HIV/AIDS and its prevention in SSA.**

| Parameter | Null model | Model 1 | Model 2 | Model 3 |
|---|---|---|---|---|
| Community-level variance | 0.980 | 0.930 | 0.882 | 0.600 |
| ICC (%) | 23.00 | 22.04 | 21.15 | 15.42 |
| PCV (%) | Reference | 5.10 | 10.00 | 38.78 |
| Deviance | 474854.22 | 468901.96 | 474001.14 | 468878 |

95%CI: 0.95–0.97) lower odds of knowledge about MTCT of HIV/AIDS and its prevention as compared to their counterparts. The odds of knowledge about MTCT of HIV/AIDS and its prevention was 1.37 (AOR = 1.37; 95%CI: 1.34–1.41), 1.40 (AOR = 1.40; 95%CI: 1.37–1.44), and 1.23 (AOR = 1.23; 95%CI: 1.19–1.28) times higher among Primiparous, multiparous, and grand multiparous mothers respectively as compared to nulliparous mothers (Table 4).

**Table 4. Multilevel analysis of factors associated with knowledge about MTCT of HIV/AIDS and its prevention among reproductive-age women in SSA.**

| Variables | Models fitted | | | |
|---|---|---|---|---|
| | Null model | Model 1 | Model 2 | Model 3 |
| | | AOR(95%CI) | AOR(95%CI) | AOR(95%CI) |
| Maternal age | | | | |
| 15–19 | | 1.00 | | 1.00 |
| 20–24 | | 1.22(1.19–1.25) | | 1.22(1.19–1.25) |
| 25–29 | | 1.31(1.27–1.35) | | 1.31(1.27–1.34) |
| 30–34 | | 1.36(1.32–1.41) | | 1.36(1.32–1.41) |
| 35–39 | | 1.40(1.36–1.45) | | 1.40(1.35–1.45) |
| 40–44 | | 1.35(1.30–1.40) | | 1.35(1.30–1.40) |
| 45–49 | | 1.27(1.22–1.32) | | 1.27(1.22–1.32) |
| Maternal education | | | | |
| No formal education | | 1.00 | | 1.00 |
| Primary | | 1.22(1.20–1.24) | | 1.22(1.20–1.25) |
| Secondary | | 1.35(1.33–1.38) | | 1.35(1.33–1.38) |
| Tertiary and higher | | 1.41(1.36–1.46) | | 1.41(1.36–1.46) |
| Current marital status | | | | |
| Married | | 0.92(0.91–0.94) | | 0.96(0.92–1.01) |
| Not married | | 1.00 | | 1.00 |
| Employment | | | | |
| Employed | | 1.02(1.01–1.03) | | 1.01(0.98–1.03) |
| Not employed | | 1.00 | | 1.00 |
| Wealth index | | | | |
| Poorest | | 1.00 | | 1.00 |
| Poorer | | 1.07(1.05–1.10) | | 1.07(1.05–1.10) |
| Middle | | 1.15(1.12–1.18) | | 1.15(1.12–1.17) |
| Richer | | 1.23(1.20–1.26) | | 1.22(1.19–1.25) |
| Richest | | 1.27(1.24–1.30) | | 1.25(1.22–1.29) |
| Mass media exposure | | | | |
| Had exposure | | 1.10(1.09–1.12) | | 1.10(1.09–1.12) |
| Not had exposure | | 1.00 | | 1.00 |
| Distance from the health facility | | | | |
| A big problem | | 0.96(0.94–0.97) | | 0.96(0.95–0.97) |

*(Continued)*

**Table 4.** (Continued)

| Variables | Models fitted | | | |
| --- | --- | --- | --- | --- |
| | Null model | Model 1 | Model 2 | Model 3 |
| | | AOR(95%CI) | AOR(95%CI) | AOR(95%CI) |
| Not a big problem | | 1.00 | | 1.00 |
| Parity | | | | |
| Nulliparous | | 1.00 | | 1.00 |
| Primiparous | | 1.37(1.34–1.41) | | 1.37(1.34–1.41) |
| Multiparous | | 1.41(1.37–1.45) | | 1.40(1.37–1.44) |
| Grand multiparous | | 1.23(1.19–1.28) | | 1.23(1.19–1.28) |
| Number of under-five children | | | | |
| None | | 1.00 | | 1.00 |
| 1–2 | | 0.99(0.98–1.01) | | 0.99(0.98–1.01) |
| 3–6 | | 0.92(0.90–0.95) | | 0.96(0.90–1.01) |
| Residence | | | | |
| Urban | | | 1.00 | 1.00 |
| Rural | | | 0.81(0.80–0.82) | 0.98(0.96–1.01) |
| Community illiteracy level | | | | |
| Low | | | 1.00 | 1.00 |
| High | | | 0.98(0.89–1.08) | 0.99(0.90–1.10) |
| Community-level of mass media non-exposure | | | | |
| Low | | | 1.00 | 1.00 |
| High | | | 1.03(0.94–1.13) | 1.09(0.99–1.21) |

## Discussion

This study aimed to assess Knowledge about maternal to child transmission of HIV/AIDS and its prevention and associated factors among reproductive-age women in SSA. In this study, 56.21% of respondents had correct knowledge about MTCT of HIV/AIDS and its prevention. This finding is lower than a study done in Zimbabwe, Tanzania, Nigeria, and the United States of America [12–14,16]. However, the finding of our study is higher than studies done in Ethiopia and Uganda [11,15,17]. This discrepancy may be because this study was based on the pooled analysis, which incorporates SSA countries. In addition, the difference in the study period and the study population might be the other reason.

In the multilevel analysis, individual-level variables such as maternal age, maternal education, wealth index, media exposure, distance from the health facility, and parity were significantly associated with knowledge about MTCT of HIV/AIDS and its prevention. Being an older age group was associated with higher odds of knowledge about MTCT of HIV/AIDS and its prevention compared to younger aged women (women aged 15–19 years). This is in line with a study done in Zimbabwe [12]. This may be attributed to the proximity of older women during their consecutive pregnancy to various maternal health services. Besides, this might indicate strategies to support younger women (adolescents) to increase awareness of HIV transmission and reduce MTCT and support ART adherence and viral suppression are not adequate [7].

Educated mothers had higher odds of knowledge about MTCT of HIV/AIDS and its prevention compared to those who had no formal education. This is in concordance with previous studies done elsewhere [11,17,26]. This may be because educated women have more access to different health-related information and can easily understand HIV/AIDS and its MTCT.

The study at hand also revealed that being from households with good socioeconomic status had higher odds of knowledge about MTCT of HIV/AIDS and its prevention as compared to their counterparts. This is in line with studies conducted in Ethiopia and Tanzania [11,27]. The greater level of awareness among women from households with good socioeconomic status could be due to their easy access to maternal health services such as PMTCT programs.

In this study, being having exposure to mass media was associated with higher odds of knowledge about MTCT of HIV/AIDS and its prevention as compared to their counterparts. This is congruent with a study done in Ethiopia [11]. This means, to eliminate MTCT of HIV, it is necessary to give special attention to illiterate and poor women in PMTCT services and touch them with targeted MTCT and PMTCT messages through various forms of mass media.

Mothers who perceived distance from the health facility as a big problem had higher odds of knowledge about MTCT of HIV/AIDS and its prevention. This may be due to women from remote areas did not have adequate access to health facilities [28], which in turn results in lower utilization of maternal health services and other infrastructures such as schooling. This results in mothers to have lower awareness about MTCT of HIV/AIDS and its prevention.

Parity was associated with knowledge about MTCT of HIV/AIDS and its prevention. Consistent with other studies [27,29], in this study, multiparous women had higher odds of knowledge about MTCT of HIV/AIDS and its prevention. This may be because multiparous women may have a greater likelihood of exposure to maternal health services, including HIV testing and counseling services, during their consecutive pregnancy.

This study was based on nationally representative data with appropriate statistical analysis (multilevel analysis). Due to this, it can help policymakers and governmental and non-governmental organizations for taking appropriate interventions. However, the study was not without limitations. First, since it was based on the information available in the survey data some factors that may be associated with the outcome variable such as quality and availability of health services and knowledge about HIV/AIDS were not assessed. Second, since it was based on the survey data we are unable to show the temporal relationship between the outcome variable and the incorporated independent variables. Furthermore, we used DHS conducted during the previous ten years, and there could be changes in awareness of MTCT and ART regimens, as well as ART uptake prior to and during pregnancy (Option B+) over time. Therefore, caution is required during the interpretation of the study results.

## Conclusion

Knowledge about MTCT of HIV/AIDS and its prevention among reproductive-age women in SSA was low. In the multilevel analysis, older age, being attending primary and above education, from rich households, having mass media exposure, perceiving distance from the health facility as not a big problem, and parous women were associated with higher odds of knowledge about MTCT of HIV/AIDS and its prevention. Therefore, it is better to consider the high-risk groups during the intervention to increase awareness about this essential public health issue and to tackle its devastating outcomes.

## Supporting information

**S1 Table. Percentage distribution of study participants by country.**
(DOCX)

**S2 Table. Prevalence of knowledge about mother to child transmission of HIV/AIDS and its prevention among reproductive age women in Sub-Saharan Africa.**
(DOCX)

## Acknowledgments

Our sincere gratitude and appreciation go to the MEASURE DHS PROGRAM for enabling us to use the data sets.

## Author Contributions

**Conceptualization:** Achamyeleh Birhanu Teshale, Zemenu Tadesse Tessema, Adugnaw Zeleke Alem, Yigizie Yeshaw, Alemneh Mekuriaw Liyew, Tesfa Sewunet Alamneh, Getayeneh Antehunegn Tesema, Misganaw Gebrie Worku.

**Data curation:** Achamyeleh Birhanu Teshale, Zemenu Tadesse Tessema, Adugnaw Zeleke Alem, Yigizie Yeshaw, Alemneh Mekuriaw Liyew, Tesfa Sewunet Alamneh, Getayeneh Antehunegn Tesema, Misganaw Gebrie Worku.

**Formal analysis:** Achamyeleh Birhanu Teshale, Zemenu Tadesse Tessema, Adugnaw Zeleke Alem, Yigizie Yeshaw, Alemneh Mekuriaw Liyew, Tesfa Sewunet Alamneh, Getayeneh Antehunegn Tesema, Misganaw Gebrie Worku.

**Investigation:** Achamyeleh Birhanu Teshale, Zemenu Tadesse Tessema, Adugnaw Zeleke Alem, Yigizie Yeshaw, Alemneh Mekuriaw Liyew, Tesfa Sewunet Alamneh, Getayeneh Antehunegn Tesema, Misganaw Gebrie Worku.

**Methodology:** Achamyeleh Birhanu Teshale, Zemenu Tadesse Tessema, Adugnaw Zeleke Alem, Yigizie Yeshaw, Alemneh Mekuriaw Liyew, Tesfa Sewunet Alamneh, Getayeneh Antehunegn Tesema, Misganaw Gebrie Worku.

**Resources:** Achamyeleh Birhanu Teshale, Zemenu Tadesse Tessema, Adugnaw Zeleke Alem, Yigizie Yeshaw, Alemneh Mekuriaw Liyew, Tesfa Sewunet Alamneh, Getayeneh Antehunegn Tesema, Misganaw Gebrie Worku.

**Software:** Achamyeleh Birhanu Teshale, Zemenu Tadesse Tessema, Adugnaw Zeleke Alem, Yigizie Yeshaw, Alemneh Mekuriaw Liyew, Tesfa Sewunet Alamneh, Getayeneh Antehunegn Tesema, Misganaw Gebrie Worku.

**Validation:** Achamyeleh Birhanu Teshale, Zemenu Tadesse Tessema, Adugnaw Zeleke Alem, Yigizie Yeshaw, Alemneh Mekuriaw Liyew, Tesfa Sewunet Alamneh, Getayeneh Antehunegn Tesema, Misganaw Gebrie Worku.

**Visualization:** Achamyeleh Birhanu Teshale, Zemenu Tadesse Tessema, Adugnaw Zeleke Alem, Yigizie Yeshaw, Alemneh Mekuriaw Liyew, Tesfa Sewunet Alamneh, Getayeneh Antehunegn Tesema, Misganaw Gebrie Worku.

**Writing – original draft:** Achamyeleh Birhanu Teshale, Zemenu Tadesse Tessema, Adugnaw Zeleke Alem, Yigizie Yeshaw, Alemneh Mekuriaw Liyew, Tesfa Sewunet Alamneh, Getayeneh Antehunegn Tesema, Misganaw Gebrie Worku.

**Writing – review & editing:** Achamyeleh Birhanu Teshale, Zemenu Tadesse Tessema, Adugnaw Zeleke Alem, Yigizie Yeshaw, Alemneh Mekuriaw Liyew, Tesfa Sewunet Alamneh, Getayeneh Antehunegn Tesema, Misganaw Gebrie Worku.

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
