## [Decision Letter · Decision Letter 0]

21 Apr 2021

PONE-D-20-37111

Knowledge about mother to child transmission of HIV/AIDS, its prevention and associated factors among reproductive-age women in sub-Saharan Africa: Evidence from 33 countries recent Demographic and Health Surveys

PLOS ONE

Dear Dr. Teshale,

Thank you for submitting your manuscript to PLOS ONE. After careful consideration, we feel that it has merit but does not fully meet PLOS ONE’s publication criteria as it currently stands. Therefore, we invite you to submit a revised version of the manuscript that addresses the points raised during the review process.

We look forward to receiving your revised manuscript.

Kind regards,

Pande Putu Januraga, M.D., DrPH

Academic Editor

PLOS ONE

Journal Requirements:

Reviewers' comments:

Reviewer's Responses to Questions

**Comments to the Author**

1. Is the manuscript technically sound, and do the data support the conclusions?

Reviewer #1: Yes

Reviewer #2: Partly

2. Has the statistical analysis been performed appropriately and rigorously? 

Reviewer #1: I Don't Know

Reviewer #2: I Don't Know

3. Have the authors made all data underlying the findings in their manuscript fully available?

Reviewer #1: Yes

Reviewer #2: Yes

4. Is the manuscript presented in an intelligible fashion and written in standard English?

Reviewer #1: Yes

Reviewer #2: Yes

5. Review Comments to the Author

Reviewer #1: Thank you for the opportunity to provide an input to this manuscript. This is a really interesting study which could fill the gap regarding PMTCT knowledge and its prevention among reproductive age women in SSA.

Minor changes to manuscript is recommended as follow:

Abstract:

Line 31: “conducted from 2008/19 to 2018/19” : is it the correct period?

Line 33: “a multilevel analysis”. Please elaborate more on this. Please also include the information regarding potential associated factors included in the analysis.

Background:

Line 59 “50 percent of the 180,000 new 60 pediatric HIV infections were infected in 2017”. What does this supposed to mean?

Line 60 : ‘it is estimated that in the absence of any 61 intervention to prevent MTCT, the risk of transmission ranges from 15-45 percent (5-10 percent 62 during pregnancy, 10-20 percent during childbirth, and 10-20 percent via mixed infant feeding).”.Can you provide a reference for this statement?

Line 64 : “Nearly 90 percent of all children and adolescents living with HIV are in Sub-Saharan Africa (SSA).”any reference for this?

Line 74: “Although the majority of the population in SSA are lived in rural areas with restricted 75 availability and accessibility of health facilities, most of the studies on knowledge about MTCT of 76 HIV/AIDS and its prevention were conducted among available women, such as those who came 77 to the health facility for their antenatal care follow up”. Can you provide a reference for this?

Also , how your study is different compare to references that you have provided on line 200 -202.

Methods

Line 84: “Demographic and Health Surveys (DHS), which were conducted from 85 2008/19 to 2018/19, was our data source”. I this the correct period?

Line 116: “weighting was done to assure the representativeness and non-response rate as well as to get an 117 appropriate statistical estimate (robust standard error)”. Can the authors elaborate more on this?

Line 120 : ‘to assess the variability of 120 the outcome between clusters”. What are the clusters?

Line 123 : “Deviance was used to verify model fitness, and the best-fit model has been 124 deemed a model with the lowest deviance”. Please mention the results of this analysis in the result section.

Results:

Line 161: “the random effect model and model fitness/comparison”. The authors did not mention the use of random effect analysis in the method section. If the authors used random effect analysis, what was the cluster being used?

Discussion:

The discussion section has been really interesting but would have been better if the author also relates the results with the current policy and programs implemented in SSA.

What are the overall results telling us about what should be done in general, to address the issues?

Reviewer #2: This manuscript evaluates knowledge of PMTCT among more than 350,000 women of reproductive age in 33 countries in SSA using DHS data. The results evaluate “comprehensive knowledge of PMTCT” based on correct responses to all four questions included in the DHS data. Overall, the majority (≥80%) of women responding to the individual questions were correct, yet combined, this was 56% on all 4 questions.

While the authors set out to use “recent DHS surveys”, some of the DHS data are fairly old (2008/09) and both knowledge of MTCT and ART regimens and uptake of ART prior to and during pregnancy (Option B+) has significantly changed compared to DHS data since 2014/15 or more recently. I would recommend the authors consider including only those countries with DHS data in the past 5-6 years to better represent recent knowledge and prevention of MTCT. This may reduce the number of countries included in the analysis, but will better reflect current knowledge, which is important.

• Lines 57-63 state that “most pregnant women are unwilling to participate in the program,...” and focuses on transmission risk during breastfeeding among women not on ART. I would restate this to reflect that in the era of Option B+ a larger number of women living with HIV (WLHIV) are on ART and while more than 70% of WLHIV are on ART in pregnancy in most SSA countries, there are still gaps to improve uptake and adherence of ART. Also important to include the importance of viral suppression and timing of ART initiation in the perinatal period as it relates to MTCT risk. Lastly, make it clear that the transmission risk during breastfeeding is among women not on ART.

• In the discussion, it would be helpful to incorporate results from studies that have tried the strategies to reduce MTCT and raise awareness to add to the statements of what should be done. For instance, adding to lines 211-213, in many countries in SSA there are targeted strategies to support younger women (adolescents) to increase awareness of HIV transmission and reduce MTCT and support ART adherence and viral suppression. Moreover, lines 225-227 regarding mass media, what have studies using mass media shown in terms of increasing awareness of HIV in the community? These data would strengthen the discussion in alignment with your results.

6. PLOS authors have the option to publish the peer review history of their article (what does this mean?). If published, this will include your full peer review and any attached files.

Reviewer #1: No

Reviewer #2: No

---

## [Author Response · Author response to Decision Letter 0]

29 Apr 2021

Date: April 29, 2021

Response to editor and reviewer comment

Title: Knowledge about mother to child transmission of HIV/AIDS, its prevention and associated factors among reproductive-age women in sub-Saharan Africa: Evidence from 33 countries recent Demographic and Health Surveys

Manuscript number: PONE-D-20-37111

Dear editor and reviewer, thank you for your comment and suggestions. We have considered all your comments and suggestions in the revised manuscript. Here, below, is the point-by-point response for the issues raised by the editor and reviewers. 

Thank you

Response to editor comment

Authors’ response: Thank you. We have confirmed that the revised manuscript meets PLOS ONE's style requirements, including those for file naming

Response to reviewers comment

Response to Reviewer #1: 

1. Abstract:

Line 31: “conducted from 2008/19 to 2018/19” : is it the correct period?

Authors’ response: Thank you very much. It was to mean 2008/09 to 2018/19 and we have corrected it in the revised manuscript.

Line 33: “a multilevel analysis”. Please elaborate more on this. Please also include the information regarding potential associated factors included in the analysis.

Authors’ response: Dear reviewer. Thank you very much for your comment. However, the detailed methodology regarding the multilevel analysis and the potential associated factors included in the analysis are found in the methods section. If we put what you have recommend in the abstract, the word of the abstract will be above 350 and this is not possible according to the PLoS ONE Journal. 

2. Background:

Line 59 “50 percent of the 180,000 new 60 pediatric HIV infections were infected in 2017”. What does this supposed to mean?

Authors’ response: Thank you. We have rewritten it to read, “About 50% of the 180,000 new pediatric HIV infections in 2017 were infected during breastfeeding”.

Line 60 : ‘it is estimated that in the absence of any intervention to prevent MTCT, the risk of transmission ranges from 15-45 percent (5-10 percent 62 during pregnancy, 10-20 percent during childbirth, and 10-20 percent via mixed infant feeding).” Can you provide a reference for this statement?

Authors’ response: Thank you. We have provided the reference. 

Line 64 : “Nearly 90 percent of all children and adolescents living with HIV are in Sub-Saharan Africa (SSA).”any reference for this?

Authors’ response: Thank you. We have provided the reference in the revised manuscript.

Line 74: “Although the majority of the population in SSA are lived in rural areas with restricted 75 availability and accessibility of health facilities, most of the studies on knowledge about MTCT of HIV/AIDS and its prevention were conducted among available women, such as those who came to the health facility for their antenatal care follow up”. Can you provide a reference for this? Also , how your study is different compare to references that you have provided on line 200 -202.

Authors’ response: Thank you. We have cited the above statement in the revised manuscript. Our study is different from other studies, especially those indicated/cited on line 200-202 since this study is based on pooled data from SSA countries. The other studies were conducted in individual countries (they represent a single country) and they may not represent the whole SSA for policymakers and for governmental and non-governmental organizations to take appropriate intervention. 

3. Methods

Line 84: “Demographic and Health Surveys (DHS), which were conducted from 85 2008/19 to 2018/19, was our data source”. Is this the correct period?

Authors’ response: Thank you. It was typing error and in the revised manuscript, we have amended to read, 2008/09 to 2018/19. 

Line 116: “weighting was done to assure the representativeness and non-response rate as well as to get an appropriate statistical estimate (robust standard error)”. Can the authors elaborate more on this?

Authors’ response: Thank you. Sampling weights are adjustment factors applied to each case in tabulations to adjust for differences in probability of selection and interview between cases in a sample. However when standard errors, confidence intervals or significance testing is required, then it is important to take into account the complex sample design of the DHS data. For the complex sample design, it is necessary to know three pieces of information – the primary sampling unit or cluster variable, the stratification variable, and the weight variable. To apply the complex sample design parameters in estimating indicators, each of the statistical software use a different set of commands and for this study we used Stata svyset and svy commands. Therefore, we have applied weighting using v005 (weight=v005/1000000) to assure representativeness and we have accounted complex sample design (using the above stata command) to get appropriate statistical estimate. Dear reviewer, we have incorporated only important advantages with reference and if we incorporate all the issues regarding weighting in the main manuscript, it may distort our method section. Therefore, in the revised manuscript, we have cited this issue in the method section for readers who need further explanation.

Line 120 : ‘to assess the variability of the outcome between clusters”. What are the clusters?

Authors’ response: Thank you. All the DHS data have cluster number represented by v001. Cluster means Enumeration areas. For example, there are 645 (202 in urban areas and 443 in rural areas) clusters/EAs for the 2016 EDHS data and overall in sub-Saharan Africa there are 1612 clusters/EAs.

Line 123 : “Deviance was used to verify model fitness, and the best-fit model has been deemed a model with the lowest deviance”. Please mention the results of this analysis in the result section.

Authors’ response: Thank you. We have included it in the random effect analysis section with bracket. Besides, we have incorporated this in table 3.

4. Results:

Line 161: “the random effect model and model fitness/comparison”. The authors did not mention the use of random effect analysis in the method section. If the authors used random effect analysis, what was the cluster being used?

Authors’ response: Thank you. We have mentioned the use of random effect analysis. we have added such information in data management and analysis section “For random effect analysis (to assess the community or cluster level variability of comprehensive knowledge towards MTCT and its prevention), ICC and proportional change in variance (PCV) was used”. Besides, the cluster being used for each country was represented by v001 and 1612 clusters were used at SSA level.

5. Discussion:

The discussion section has been really interesting but would have been better if the author also relates the results with the current policy and programs implemented in SSA.

Authors’ response: Than you. It is difficult to relate/discuss each finding with the policy and programs what exist before (there may not be program before). However, we have tried to relate our findings with current policy and programs in the revised manuscript. Moreover, when concluding our results, we have concluded to the standard of policymakers and responsible bodies in the area. 

What are the overall results telling us about what should be done in general, to address the issues?

Authors’ response: Thank you. The overall result tells that ONLY 56.21% of mothers in SSA had comprehensive knowledge AND to increase maternal knowledge towards MTCT and PMTCT; interventions should be targeted to factors at individual level such as those with no formal education, those from remote areas, low socioeconomic status, nulliparous mothers, and those mothers with no access to media.

Response to Reviewer #2:

1. While the authors set out to use “recent DHS surveys”, some of the DHS data are fairly old (2008/09) and both knowledge of MTCT and ART regimens and uptake of ART prior to and during pregnancy (Option B+) has significantly changed compared to DHS data since 2014/15 or more recently. I would recommend the authors consider including only those countries with DHS data in the past 5-6 years to better represent recent knowledge and prevention of MTCT. This may reduce the number of countries included in the analysis, but will better reflect current knowledge, which is important.

Authors’ response: Thank you very much. It is difficult to conclude the findings from four or five countries in SSA to the whole SSA countries. Therefore, we prefer to conduct this study using DHS conducted in the past 10 years, despite its limitations. Dear reviewer, we have acknowledged your issue as limitation of this study in the last paragraph of the discussion section. 

2. Lines 57-63 state that “most pregnant women are unwilling to participate in the program,...” and focuses on transmission risk during breastfeeding among women not on ART. I would restate this to reflect that in the era of Option B+ a larger number of women living with HIV (WLHIV) are on ART and while more than 70% of WLHIV are on ART in pregnancy in most SSA countries, there are still gaps to improve uptake and adherence of ART. Also important to include the importance of viral suppression and timing of ART initiation in the perinatal period as it relates to MTCT risk. Lastly, make it clear that the transmission risk during breastfeeding is among women not on ART.

Authors’ response: Thank you. We have accepted your suggestions and comments and we have incorporated them in the revised manuscript.

3. In the discussion, it would be helpful to incorporate results from studies that have tried the strategies to reduce MTCT and raise awareness to add to the statements of what should be done. For instance, adding to lines 211-213, in many countries in SSA there are targeted strategies to support younger women (adolescents) to increase awareness of HIV transmission and reduce MTCT and support ART adherence and viral suppression. Moreover, lines 225-227 regarding mass media, what have studies using mass media shown in terms of increasing awareness of HIV in the community? These data would strengthen the discussion in alignment with your results.

Authors’ response: Thank you. We have considered your comments accordingly.

---

## [Editor Report · Decision Letter 1]

27 May 2021

PONE-D-20-37111R1

Knowledge about mother to child transmission of HIV/AIDS, its prevention and associated factors among reproductive-age women in sub-Saharan Africa: Evidence from 33 countries recent Demographic and Health Surveys

PLOS ONE

Dear Dr. Teshale,

Thank you for submitting your manuscript to PLOS ONE. After careful consideration, we feel that it has merit but does not fully meet PLOS ONE’s publication criteria as it currently stands. Therefore, we invite you to submit a revised version of the manuscript that addresses the points raised during the review process.

We look forward to receiving your revised manuscript.

Kind regards,

Pande Putu Januraga, M.D., DrPH

Academic Editor

PLOS ONE

Journal Requirements:

Additional Editor Comments (if provided):

Dear Authors,

Thank you for submitting the revised version of the manuscript with responses to the reviewers.

I believe that the authors have responded to the reviewers' comments appropriately. However, I still have minor comments that need to be responded to before a final decision.

Editor comments:

Line 57, for some readers, particularly in non-generalized epidemics context, the option B+ may be difficult to understand; please provide an explanation for the term.

Line 72-74 is only a one-sentence paragraph; please integrate the line into the next paragraph.

All the best,

Pande

---

## [Author Response · Author response to Decision Letter 1]

29 May 2021

Date: May 28, 2021

Rebuttal letter 

Title: Knowledge about mother to child transmission of HIV/AIDS, its prevention and associated factors among reproductive-age women in sub-Saharan Africa: Evidence from 33 countries recent Demographic and Health Surveys

Manuscript number: PONE-D-20-37111R1

Dear editors and reviewers, thank you for your constructive comments and suggestions. Here, below, is the authors’ point-by-point response for the comments raised. 

Response to questions regarding Journal Requirements

Authors’ Response: Thank you for the comment. The author have confirmed that there is no retracted article in the reference list and we have checked and rewritten reference 1 and 2 accordingly (see the revised manuscript). 

Response for Additional Editor Comments 

Line 57, for some readers, particularly in non-generalized epidemics context, the option B+ may be difficult to understand; please provide an explanation for the term.

Authors’ Response: Option B+ is offering immediate antiretroviral therapy for mothers living with HIV to prevent the vertical transmission HIV/AIDS regardless of their CD4 count. That is, option B+ is initiation of antiretroviral therapy for all pregnant mothers, regardless of their CD4 count, to PMTCT of HIV/AIDS. In the revised manuscript, we have consider such points.

Line 72-74 is only a one-sentence paragraph; please integrate the line into the next paragraph.

Authors’ Response: Thank you. We have considered your comment (we have added the lines into the next paragraph) in the revised manuscript.

---

## [Editor Report · Decision Letter 2]

1 Jun 2021

Knowledge about mother to child transmission of HIV/AIDS, its prevention and associated factors among reproductive-age women in sub-Saharan Africa: Evidence from 33 countries recent Demographic and Health Surveys

PONE-D-20-37111R2

Dear Dr. Teshale,

We’re pleased to inform you that your manuscript has been judged scientifically suitable for publication and will be formally accepted for publication once it meets all outstanding technical requirements.

Kind regards,

Pande Putu Januraga, M.D., DrPH

Academic Editor

PLOS ONE
---

## [Editor Report · Acceptance letter]

3 Jun 2021

PONE-D-20-37111R2 

Knowledge about mother to child transmission of HIV/AIDS,  its prevention and associated factors among reproductive-age women in sub-Saharan Africa: Evidence from 33 countries recent Demographic and Health Surveys 

Dear Dr. Teshale:

I'm pleased to inform you that your manuscript has been deemed suitable for publication in PLOS ONE. Congratulations! Your manuscript is now with our production department. 

Kind regards, 

on behalf of

Dr. Pande Putu Januraga 

Academic Editor

PLOS ONE